# Principles for enhancing virus capsid capacity and stability from a thermophilic virus capsid structure

Nicholas P. Stone [1], Gabriel Demo [2], Emily Agnello [3] & Brian A. Kelch [1]*

The capsids of double-stranded DNA viruses protect the viral genome from the harsh extracellular environment, while maintaining stability against the high internal pressure of packaged DNA. To elucidate how capsids maintain stability in an extreme environment, we use cryoelectron microscopy to determine the capsid structure of thermostable phage P74-26 to 2.8-Å resolution. We find P74-26 capsids exhibit an overall architecture very similar to those of other tailed bacteriophages, allowing us to directly compare structures to derive the structural basis for enhanced stability. Our structure reveals lasso-like interactions that appear to function like catch bonds. This architecture allows the capsid to expand during genome packaging, yet maintain structural stability. The P74-26 capsid has T = 7 geometry despite being twice as large as mesophilic homologs. Capsid capacity is increased with a larger, flatter major capsid protein. Given these results, we predict decreased icosahedral complexity (i.e. T ≤ 7) leads to a more stable capsid assembly.

[1] Department of Biochemistry and Molecular Pharmacology, University of Massachusetts Medical School, Worcester, MA 01655, USA. [2] RNA Therapeutics Institute, University of Massachusetts Medical School, Worcester, MA 01655, USA. [3] Graduate School in Biomedical Sciences, University of Massachusetts Medical School, Worcester, MA 01655, USA. *email: brian.kelch@umassmed.edu

Capsids are protein shells that surround and protect the viral genome. Capsid proteins often self-assemble into icosahedral structures with a quasi-equivalent arrangement of individual subunits. In quasi-equivalence theory[1], the capsid subunits use similar interactions throughout the assembly, but are arranged in slightly different ways in non-symmetry related environments.

Icosahedral capsid proteins form substructures called capsomers, with each capsomer comprising either five or six subunits (pentons and hexons, respectively). Typical icosahedral capsids can be described by the triangulation number ($T$), which describes the complexity of the icosahedral symmetry. Capsids are composed of 60 $T$ protein subunits, resulting in an assembly of 12 pentons and a variable number of hexons ($10 \times (T-1)$). Although capsid structure has been intensely studied for over a half century, the principles underlying capsid size and stability remain elusive.

Capsid size is typically controlled by the underlying capsid geometry: the triangulation number and whether the capsid is prolate or isometric. Most frequently, the triangulation number is altered to control capsid size. Higher triangulation number results in more hexons and, therefore, a larger capsid; lowering the triangulation number will result in a smaller capsid. Another common mechanism for increasing capsid capacity is to convert a spherical, isometric capsid into an elongated icosahedron, referred to as a prolate capsid. A prolate capsid consists of a cylinder with two quasi-icosahedral caps at the ends, resulting in increased interior capacity. In many viruses, an interior scaffolding protein or domain dictates the assembly of homogeneous capsids with a defined geometry and size[2,3]. Small alterations to capsid protein primary sequence can change T number or can convert a capsid between isometric and prolate[4,5]. Thus, the evolutionary barriers for altering capsid size through these mechanisms are quite low. It remains unknown if there are other mechanisms for capsid size control and whether these mechanisms have any ramifications for virus fitness, such as capsid stability.

Capsid stability is vital for virion survival in an external fluctuating environment until it finds a new host cell. Across the viral world, there is a wide spectrum of capsid stabilities[6]. Tailed phages and similar viruses utilize more stable capsids because of the high internal pressure of their tightly packaged genome[7,8] and because the capsid never disassembles during the infection cycle[9–11]. Tailed bacteriophages, as well as herpesviruses, use Major Capsid Proteins of the conserved HK97 fold[12]. Capsids of these viruses first form as spherical procapsids, which convert to an icosahedral shape concomitant with genome packaging, release, or cleavage of auxiliary proteins. The conversion to an icosahedral shape is mediated through conformational rearrangement of the HK97 folds[2,13].

Thermophilic virus capsids are among the strongest because they survive in an especially harsh environment. Previous studies of thermophilic viruses have focused on capsids of various shapes including icosahedral[14], filamentous[15–17], helical[18], and lemon-shaped[19,20]. However, for most of these viruses, close mesophilic homologs are not available, which makes it challenging to identify the structural mechanisms that underlie thermostability.

We use the thermophilic, tailed bacteriophage P74-26 to elucidate the structural mechanisms of thermostability[21–23]. P74-26 is an especially long-tailed siphovirus that infects *Thermus thermophilus*[24,25] (Fig. 1a). We use P74-26 to compare with similar lambdoid phages and other mesophilic Caudoviruses. Because stability mechanisms of lambdoid phages have been studied in great detail[26–30], this comparison presents a unique opportunity to identify mechanisms of capsid stability.

Here, we report the structure of the P74-26 capsid. A series of lassos topologically tether subunits together to stabilize the capsid. The decoration protein forms a unique cage to lock the mature capsid in place. Finally, the structure reveals $T = 7$ geometry despite a capsid capacity about twice that of typical $T = 7$ Caudoviruses. P74-26 uses larger and flatter capsomers to achieve this larger capsid. Our work suggests that capsid geometry plays a critical role in virus stability.

## Results

**Thermostability of P74-26 virions.** We sought to elucidate the principles underlying the thermal stability of the P74-26 virus particle. Although it is clear that P74-26 is more stable than its mesophilic counterparts, the thermostability of P74-26 particles has never been directly measured. To address this, we heated samples of P74-26 virions to 80 °C and measured phage infectivity as a proxy of virus particle integrity (Fig. 1b). This is a very stringent test for capsid stability, as disruption of many other structures within the virion (e.g. tail, neck, baseplate, etc.) can lead to loss of infectivity. Indeed, we find that P74-26 capsids remain intact after loss of tails and packaged DNA following incubation at 80 °C (Supplementary Fig. 1). P74-26 virions remain stable and infectious when incubated at 80 °C for long periods ($t_{1/2} = 49.5 \pm 0.7$ min; three replicates). In contrast, mesophilic lambda phage rapidly loses infectivity when incubated at high temperatures in a similar buffer ($t_{1/2} = 48$ s at 75 °C and extrapolated $t_{1/2}$ ~5–7 s at 80 °C[26]). These experiments demonstrate that P74-26 is, to our knowledge, the most stable Caudovirus known.

**Overall capsid structure.** To determine the structural mechanism of P74-26 stability, we used single particle cryo-EM to determine the high-resolution structure of the P74-26 capsid. In the raw images, the capsids are clearly isometric, icosahedral particles filled with DNA (Fig. 2a, b). The ~0.9 μm long, flexible tails emanate from a fivefold vertex. We extracted capsid particles from the raw images and performed single particle reconstruction imposing icosahedral symmetry (Supplementary Fig. 2). Because the portal and tail complex sit at a unique vertex, icosahedral averaging removes features for these structures. The resolution of the reconstruction is 2.8 or 3.1 Å according to gold standard 0.143 or 0.5 Fourier Shell Correlation (FSC) criteria (Fig. 2c; Supplementary Fig. 3a, b). (However, we note that the disordered DNA in the interior can adversely impact these calculations.) The P74-26 capsid is 824 Å in diameter from vertex to vertex and 770 Å from face to face. The capsid exhibits $T = 7$ (*laevo*) symmetry, which is surprising for a capsid of this size (Fig. 2b). We could easily identify density consistent with the major capsid protein (gene product 86, hereafter referred to as MCP) and the decoration protein (gene product 87, hereafter referred to as Dec$^{P74-26}$). We had previously determined the crystal structure of the trimeric globular regions of Dec$^{P74-26}$, which consists of a β-tulip domain followed by a mixed α/β subdomain (Fig. 3a; Supplementary Fig. 3c)[23]. The Dec$^{P74-26}$ crystal structure was easily placed into the cryo-EM maps with minor adjustments. Furthermore, we clearly resolve the entire ~23 residue N-terminal arm of Dec$^{P74-26}$ (hereafter called the Dec-arm), most of which was missing from the crystal structure. Likewise, the entire chain of MCP is clearly represented in the reconstruction maps. Thus, we have determined the complete structures, from N- to C-termini, of the two major components of the P74-26 capsid (see Supplementary Table 1 for reconstruction and model statistics).

MCP adopts the expected HK97 fold (Fig. 3b; Supplementary Figs. 3d and 4). The HK97 fold has two globular domains: the rectangular P-domain (peripheral domain; residues 107–186, and 326–373) and the triangle-shaped A-domain (axial domain; residues 192–320, and 384–391). Attached to these domains are a

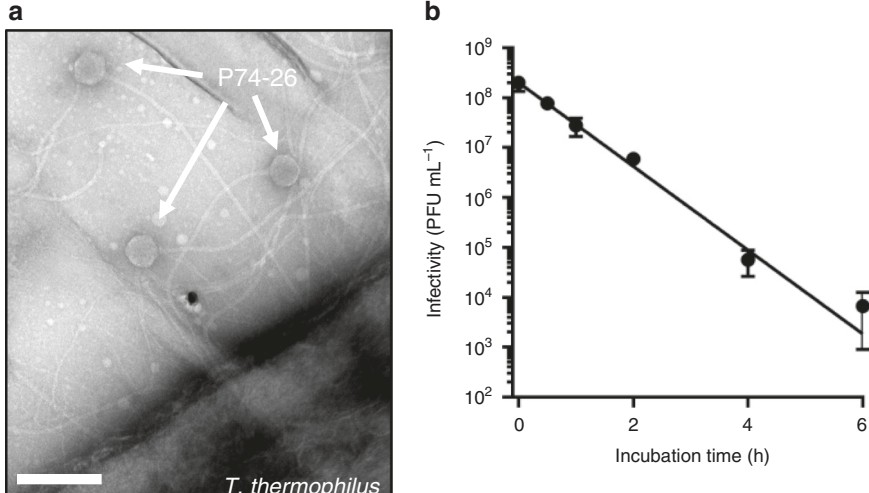

**Fig. 1** P74-26 is a thermostable virus. **a** Negative stain electron micrograph of phage P74-26 infecting *T. thermophilus* HB8; scale bar, 200 nm. **b** Purified P74-26 virions retain significant infectivity after 80 °C incubation; error bars = standard deviation calculated from $n = 3$ independent experiments. Source data for this experiment are provided as a Source Data file

series of loops and extended arms that facilitate protein–protein interactions to stabilize the capsid. MCP contains an especially long N-terminal region (the N-arm; residues 1–52) and β-hairpin called the E-loop (extended loop; residues 60–106). MCP also contains four non-classical elements that are either unique to P74-26 or are not found in most HK97 folds: (1) a G-loop region (residues 166–188) of the P-domain that forms a small beta-hairpin; (2) an S-loop (residues 123–133) that lies at the N-terminal end of the P-domain spine helix; (3) a T-loop (residues 330–346) that forms a flap off the bottom of the P-domain; and (4) an extended C-terminal arm (the C-arm; residues 392–409) that extends away from the A-domain.

We observe conformational heterogeneity among the seven copies of MCP in the asymmetric unit that is restricted to the E-loop and N-arm (Supplementary Fig. 5a). The MCP conformer found in the penton exhibits larger conformational changes than the hexon subunits. The E-loop displays the largest variability in conformation with a ~27 Å movement of the tip (comparing the penton subunit to the D subunit of the hexon). The N-arm is the only other region that displays large heterogeneity across MCP conformers, with a ~19 Å movement of the N-arm tip. Both conformational changes can be described as rigid body rotations at hinge regions that lie at the bases of the E-loop and N-arm. There are minimal conformational differences throughout the rest of the HK97 fold. This is different than observed in other Caudoviruses such as phage Sf6, where there is conformational variability throughout the HK97 fold[31].

**The P74-26 capsid is exceptionally large for a $T = 7$ virus.** To our knowledge, the P74-26 capsid is far larger than observed in all other structures of $T = 7$ viruses. Compared to other $T = 7$ phage, the P74-26 capsid inner diameter is longer by ~140 Å than average, and 115 Å longer than the next biggest capsid (Fig. 4a; Supplementary Table 2). This results in a capsid capacity that is about twice as large as normal for $T = 7$ Caudoviruses. The large capsid holds the 83 kb genome of P74-26, which is nearly twice as long as average for $T = 7$ Caudoviruses. The coevolution of larger capacity and genome results in a typical packaging density (0.52 vs. 0.54 bp nm$^{-3}$ for P74-26 and other Caudoviruses, respectively).

The larger size of the $T = 7$ capsid is due to much larger and flatter capsomers than found in other Caudoviruses. The P74-26

pentons and hexons are 157 and 197 Å in diameter in the longest dimension, larger than normal capsomers (average of 129 and 158 Å, respectively). The P74-26 capsomers are also less curved than found in typical Caudoviruses, even though the spherical factor of the P74-26 capsid particle is higher than homologs (approximately 0.4 for P74-26 vs. 0.25 for HK97) (Fig. 4b; Supplementary Fig. 5b, c).

P74-26 MCP occupies much more surface area than observed in other HK97 folds. The MCP of P74-26 contains a typical number of residues (Supplementary Table 3), so this is not simply due to lengthening the MCP primary sequence. Instead, P74-26 efficiently uses extended loop architecture to create a structure with large surface area. There are eight structural changes that increase the surface area of MCP. The first change is the P74-26 E-loop β-hairpin, which is 44 residues in length, whereas the range of E-loops in other HK97 folds is 32–34 residues. The greater number of residues allows the E-loop to extend much further (~70 vs. ~53 Å for P74-26 and HK97 phage, respectively). The second change is an elongated N-arm (52 residues extending ~44 Å vs. 36 residues and ~37 Å extension, averaged across other Caudoviruses). The N-arm of P74-26 also reverses direction to directly contact the P-domain. This unique N-arm architecture covers much more surface area than other N-arms. The third change is the P74-26 C-arm, which forms a unique extended structure that runs along the outside of the A-domain. The C-arm binds two adjacent A-domains within a capsomer such that the axial region of the capsomer is enlarged (Fig. 2d, e; Supplementary Fig. 6a). The A-domain surface area is increased by the fourth and fifth major structural changes: a unique helix at the tip of the A-domain that extends the axial dimension by ~6 Å, and an extended loop region (residues 295–311) that widens the A-domain (Fig. 4c, d). The final three major structural changes are three loops that emanate from the P-domain (the G-loop, S-loop, and T-loop; Fig. 4d). Each of these three regions of the P-domain play critical roles in intra- or inter-capsomer interactions, which we delineate below.

**MCP rings, lassos, and flaps topologically link subunits.** The MCP–MCP interactions within a capsomer are much more extensive and intricate than in other Caudoviruses (Fig. 5a). The interaction area between two adjacent MCP proteins is ~3150 Å$^2$, which is larger than most other $T = 7$ HK97 folds. Only phage Sf6

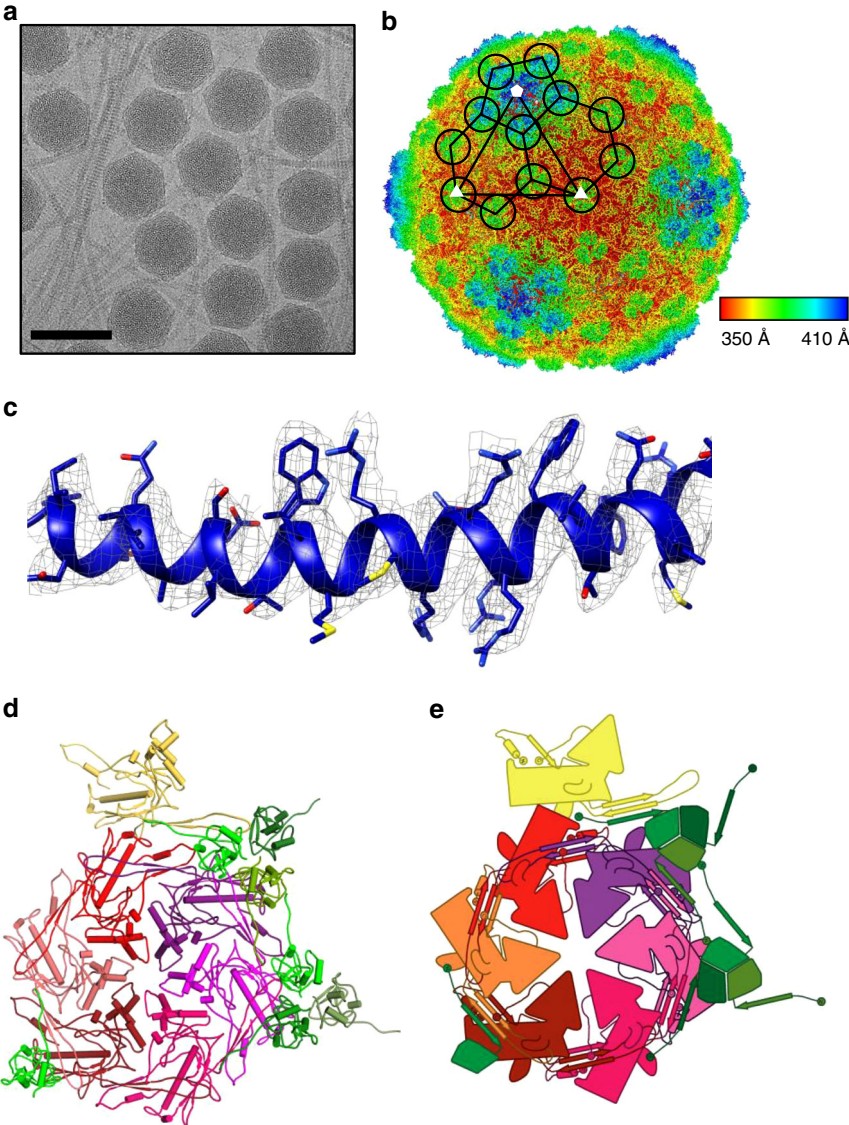

**Fig. 2** Determination of P74-26 capsid structure by cryo-EM. **a** Representative micrograph of purified P74-26 virions from high-resolution data collection; scale bar, 100 nm. **b** Icosahedral reconstruction of P74-26 mature capsid, colored radially from the center of the volume (red to blue). One penton and two hexons of MCP are outlined on the volume, and decoration protein trimers are circled. The black triangle indicates the icosahedral asymmetric unit, with three- and fivefold axes labeled as small white triangles and pentagons, respectively. **c** Representative section of the density map with fitted model of the MCP P-domain spine helix. **d**, **e** Atomic model of P74-26 icosahedral asymmetric unit (**d**) and corresponding cartoon schematic (**e**). In the asymmetric unit, the Major Capsid Protein (MCP) comprises one complete hexon (shown in shades of red) and a single penton subunit (yellow). The Decoration protein subunits (gp87) are shown in shades of green

buries more surface area (~3277 Å$^2$), much of which is contributed by the large insertion domain that makes intra-capsomer interactions in lieu of a decoration protein[31]. The hydrophobic contribution to interface stability is greater for P74-26 than other $T = 7$ Caudoviruses. The estimated hydrophobic interaction free energy between two MCP subunits is $-34.1$ kcal mol$^{-1}$ for P74-26 vs. an average of $-25.0$ kcal mol$^{-1}$ for other viruses (Supplementary Table 3). However, the number of hydrogen bonds and salt bridges stabilizing intra-capsomer interactions are typical (34 vs. 31 hydrogen bonds, and 4 vs. 7 salt bridges for P74-26 and others, respectively).

The architecture of the interfaces between MCP subunits is substantially different for P74-26, with loops and extended arms that are intertwined to provide stability against internal pressure. Like other HK97 folds, the E-loop forms important subunit–subunit interactions within the capsomer. However, there are two unique

networks of interactions that form topological links to establish tight E-loop binding.

The Dec-arm, N-arm, and C-arms all collaborate to form a ring that completely surrounds the β-sheet region of the E-loop. The Dec-arm forms an anti-parallel β-sheet with the E-loop of a neighbor, forming the outer surface of the capsid (Fig. 5a), similar to that of phage lambda[29]. The N-arm consists of four sections: (1) a hand that comprises the N-terminal four residues, (2) a helical forearm, (3) a turn that forms the elbow or crook, and (4) the upper arm that forms a β-strand conformation. The hand, forearm, and upper arm all form intimate contacts with the E-loop of an adjacent MCP. The upper arm forms a parallel β-sheet with the E-loop along the inner surface of the capsid shell, similar to that observed in nearly all other HK97 folds[12]. However, the P74-26 N-arm is unique in that the elbow interacts with a neighboring Dec protein (see below for more details) and makes a

sharp turn to orient the forearm and hand underneath the E-loop β-sheet. The forearm helix and hand interact with

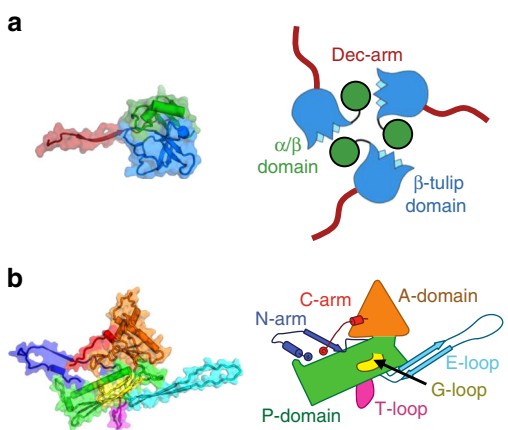

**Fig. 3** Structural models of P74-26 MCP and Dec. **a** Structure of P74-26 Dec protein monomer colored by domain (left) with corresponding schematic of the Dec trimer (right). **b** Structure of P74-26 MCP colored by domain (left) with corresponding labeled schematic (right)

the E-loop using tightly packed hydrophobic interactions, with the N-terminus amine group forming salt bridges with the P-domain (Glu142) and the E-loop of the neighbor (Asp72) (Fig. 5c).

P74-26 MCP contains a C-arm that stabilizes inter-subunit interactions and, to our knowledge, is completely absent in other HK97 folds. The 18-residue C-arm extends down the length of the A-domain, ending at the E-loop of the neighboring MCP. The C-arm uses hydrophobic interactions to glue the A-domains of two adjacent MCP subunits together. The C-terminal four residues bind directly on top of the E-loop of a neighbor, interacting with all four strands in the sheet (Fig. 5d). The four side chains form a series of hydrophobic and salt bridge interactions to stabilize the C-arm over the sheet, with the C-terminal carboxylate forming salt bridges with Arg12 of the Dec-arm and Lys97 of the E-loop. Thus, both N- and C-arms grip the neighboring E-loop in a pincer architecture, with the C-arm binding from above, while the N-arm binds from below (Supplementary Fig. 6b, c). All three arms act to completely surround the E-loop of each neighbor in a ring of tight interactions, stabilizing the inter-subunit assembly (Fig. 5e). To our knowledge, this type of architecture is unique.

The end of the E-loop attaches to the neighboring MCP. The E-loop is intercalated by several residues from the neighboring

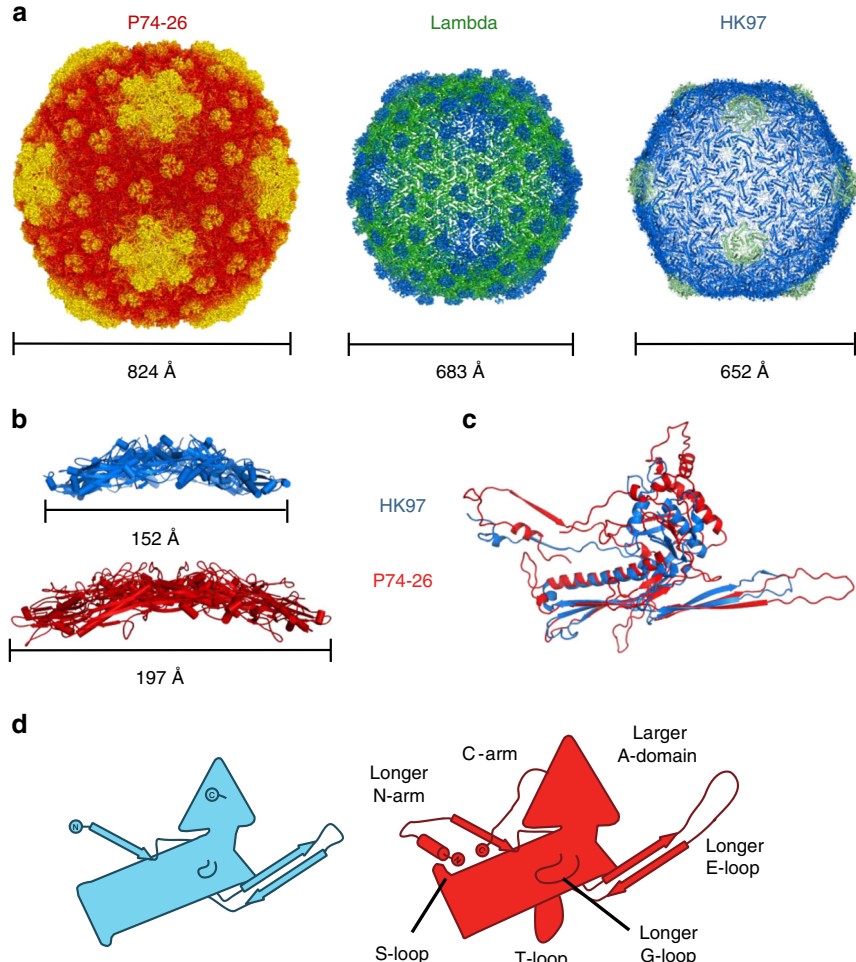

**Fig. 4** P74-26 has significantly increased capsid capacity compared to $T = 7$ mesophilic viruses. **a** Size comparison of P74-26 and homologous mesophilic $T = 7$ phages Lambda and HK97. Corresponding capsid diameters (longest outer diameter measured from fivefold axes) are listed below each structure. **b** Size comparison of hexons from HK97 (blue) and P74-26 (red); longest hexon diameters are listed below each structure. **c** Overlay of HK97 and P74-26 MCP, aligned on the HK97 fold P-domain. **d** Schematic representations of MCP from HK97 (left, blue) and P74-26 (right, red). Labels indicate features unique to P74-26 MCP

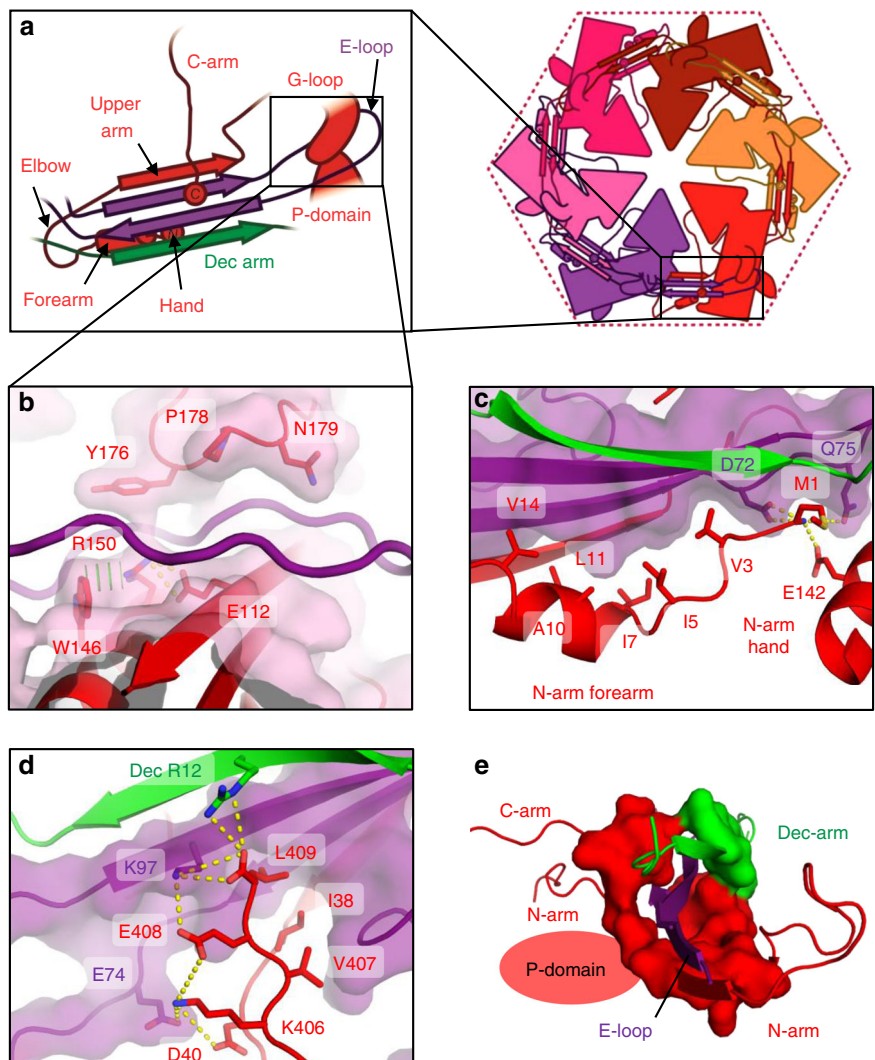

**Fig. 5** Lasso interactions stabilizing the E-loop. **a** Diagrams depicting interactions of the MCP E-loop (purple) with a neighboring MCP subunit (red) and Dec-arm (green). **b** The E-loop lasso (purple) is stabilized by the G-loop and P-domain of a neighboring subunit (red). **c** The N-arm forearm and hand bind underneath an adjacent E-loop. **d** The C-arm binds on top of all four strands of the E-loop β-sheet. **e** The β-hairpin of the E-loop (purple) is stabilized by a closed ring consisting of the MCP N- and C-arms (red), and the Dec-arm (green)

subunit (Glu112, Trp146, and Arg150 of the P-domain and Tyr176, Pro178, and Asn179 of the G-loop), preventing it from forming a canonical beta-hairpin structure (Fig. 5b; Supplementary Fig. 6d). The long G-loop is critical for this interaction, as it allows these residues to interdigitate between the E-loop strands. The interaction between the P-domain and G-loop of one subunit with the E-loop of its neighbor is mediated by numerous hydrophobic interactions, hydrogen bonds, and salt bridges (Supplementary Fig. 6d). In this manner, the E-loop acts as a lasso to topologically attach to the next subunit within the capsomer.

The N-arm also topologically links multiple proteins within the capsid assembly (Fig. 6a). The elbow interacts tightly with the P-domain of a neighboring MCP, and the β-tulip domain and Dec-arm of a nearby Dec[P74-26] (Fig. 6b). The elbow-to-Dec interaction is critical for stabilizing inter-capsomer interactions (described in more detail in the following section). The curve of the elbow directs the forearm and hand back toward the same HK97 fold from which the N-arm emanates (Fig. 6a). The P-domain contains the S-loop, a small loop of 11 residues that inserts into a groove between the forearm helix and the hand residues. These structural elements are locked together using a complicated web

of hydrogen bonds, salt bridges, and hydrophobic interactions that latches the forearm and hand in place (Fig. 6c). In this manner, the S-loop effectively closes the N-arm into a long, closed loop that links different capsid subunits together. Taken together, we observe two separate lasso interactions emanating from each side of MCP: one consisting of the E-loop and the other comprising the N-arm/S-loop combination.

The N-arm forms overlapping flaps that stabilize the assembly at the twofold and quasi-twofold axes of symmetry between capsomers (Fig. 7a, c; Supplementary Fig. 7a). The N-arm elbow region of an MCP subunit in one capsomer interacts with the S-loop, forearm, and hand of the MCP subunit in the capsomer across the twofold/quasi-twofold axis. The interaction is primarily mediated through hydrophobic interactions. Importantly, the sidechain of Leu127 in the S-loop sits inside the opening of N-arm such that it acts as a hitching post that the N-arm wraps around (Supplementary Fig. 7a). The elbow is placed on the outside face of the capsid, while the forearm and hand form the inside surface of a neighboring capsomer. Thus, both MCP proteins form two inter-locking flaps across the twofold and quasi-twofold axes.

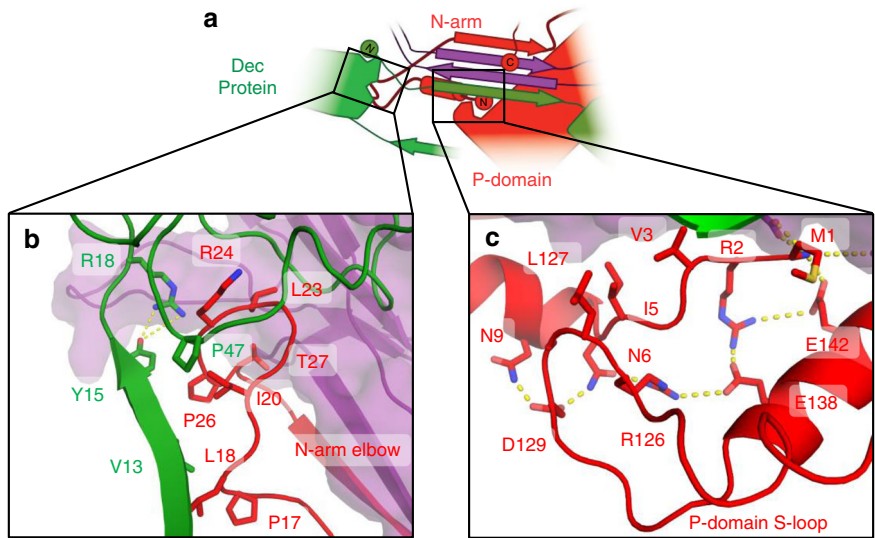

**Fig. 6** Lasso interactions stabilizing the N-arm. **a** Diagram depicting the N-arm lasso interactions with the P-domain (red) and adjacent Dec[P74-26] protein (green). **b** The N-arm elbow (red) extends to the β-tulip domain of a Dec[P74-26] protein (green) at an adjacent threefold axis. **c** The N-arm hand binds to the P-domain S-loop, creating a closed loop lasso

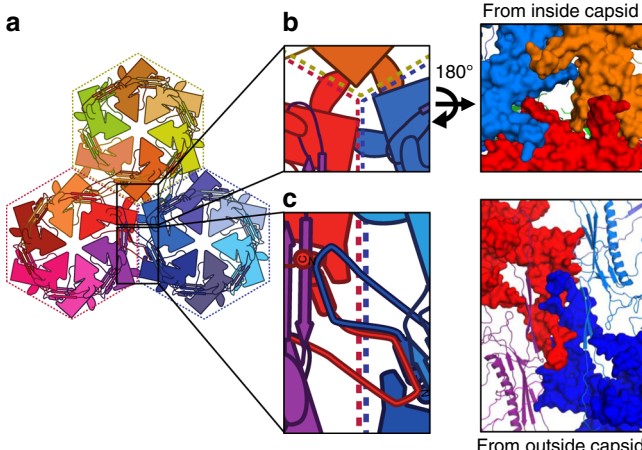

**Fig. 7** Inter-capsomer stabilization by extended flaps in P74-26 MCP. **a** Diagram of the threefold axis where three capsomers (α-red, β-blue, and γ-yellow) meet. **b** The T-loop flap interacts with the P-domain of a neighboring subunit, forming a ring around the quasi-threefold/threefold axes. Right, tongue-in-groove interactions of T-loops and adjacent P-domains, as viewed from the capsid interior. **c** The N-arms of two adjacent subunits form interleaving flaps where the N-arm binds to the S-loop of the neighboring subunit across the quasi-twofold/twofold axes

MCP uses the T-loop to stabilize the interaction where three capsomers meet (i.e. the threefold and quasi-threefold axes) (Fig. 7a). (We define these capsomers as α, β, and γ for clarity.) The T-loop forms the inside surface of the capsid, where it binds the P-domain of an MCP in a neighboring capsomer in a tongue-in-groove fashion (Fig. 7b; Supplementary Fig. 7b). The interactions are reciprocated such that the T-loop of capsomer α binds the P-domain of capsomer β, and so on. Thus, the elbows form twofold flaps on the capsid exterior to stabilize the twofold and quasi-twofold axes, while the T-loop forms tongue-in-groove interactions on the capsid interior to stabilize the threefold and quasi-threefold axes.

**The decoration protein forms an inter-capsomer cage.** The Dec[P74-26] trimer binds the capsid at the icosahedral threefold and

quasi-threefold axes, with the Dec-arm pointing to the neighboring threefold/quasi-threefold axis. Dec[P74-26] forms numerous interactions throughout the capsid. Each Dec[P74-26] subunit contacts nine different proteins: six MCP subunits in three different capsomers, two other Dec[P74-26] subunits within the same trimer, and one Dec[P74-26] subunit at the neighboring threefold/quasi-threefold axis (Fig. 8a). The total interaction area for a single 146-residue Dec[P74-26] subunit is ~4100 Å². This is vastly more extensive than seen for mesophilic decoration proteins. For example, phage TW1 has a decoration protein of similar size (149 residues), but each Dec[TW1] subunit buries only ~60% of the equivalent surface area (~2670 Å²) and interacts with seven other protein chains. Additionally, Dec[P74-26] uses much greater hydrophobic interactions than mesophilic homologs (estimated free energy of hydrophobic interactions is −38 vs. −9 kcal mol⁻¹ for P74-26 and TW1, respectively; Supplementary Table 4).

The majority of Dec[P74-26] interactions are with capsomer α, where it contacts two separate MCP chains, termed α1 and α2. The Dec[P74-26] globular region primarily makes contact with the P-domain and G-loop of the α1 subunit. The Dec[P74-26] to P-domain interaction surface comprises a series of hydrophobic interactions and salt bridges (Supplementary Fig. 8a). The interface with the α2 subunit is the most substantial, accounting for ~1300 Å² of interaction area. Most of this interaction is due to the previously mentioned interaction of the Dec-arm with the E-loop (Fig. 6a). However, additional contacts are formed between the Dec-arm and the α2 G-loop (Supplementary Fig. 8b).

The Dec-arm reaches to the Dec[P74-26] trimer at the neighboring threefold/quasi-threefold axis. The N-terminal seven residues of the Dec-arm use hydrogen bonds and salt bridges to bind the neighboring Dec[P74-26] β-tulip domain in an induced-fit mechanism (Fig. 8c; Supplementary Fig. 9). This same Dec subunit reciprocates this interaction using its Dec-arm, to create two proteins embracing each other at arm's length. These interactions effectively link each of the Dec[P74-26] trimers throughout the capsid surface into one interlocked cage (Fig. 8b). To our knowledge, this interconnected architecture of decoration proteins has not been observed in mesophilic viruses.

Each Dec[P74-26] globular region also interacts with the other two neighboring capsomers, contacting three MCP subunits in capsomer β and one in capsomer γ. The β1 MCP subunit uses its

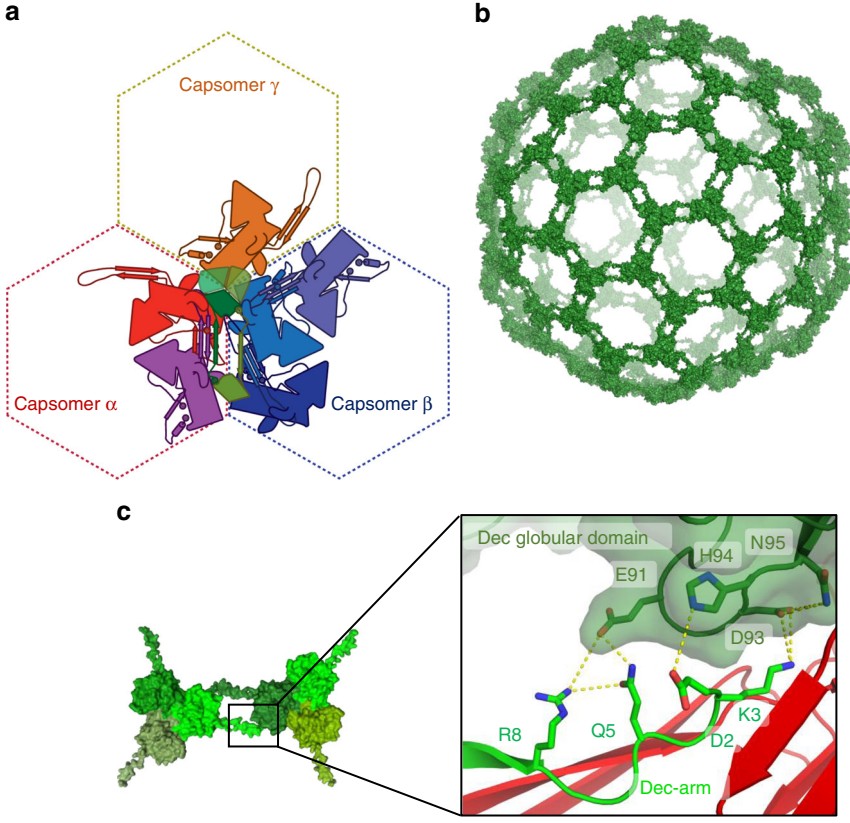

**Fig. 8** Stabilization of the P74-26 capsid by Dec[P74-26]. **a** A single Dec[P74-26] protein monomer (dark green) stabilizes capsomer interfaces by binding nine different proteins: three Dec[P74-26] subunits (green) and six MCP subunits from three capsomers. **b** The Dec[P74-26] proteins form an interconnected cage that surrounds the capsid. **c** The Dec-arm from one subunit (light green) interacts with the β-tulip domain of a Dec[P74-26] protein (dark green) positioned at an adjacent quasi-threefold/threefold axis

elbow region to bind Dec[P74-26], burying 449 Å² of area. The G-loop of the β2 MCP subunit binds the side of the Dec[P74-26] β-tulip domain, using hydrogen bonds and hydrophobic interactions, burying 257 Å² (Supplementary Fig. 10a). The β3 MCP subunit binds underneath the Dec[P74-26] globular region, using the tip of the E-loop to make hydrogen bonds (Supplementary Fig. 10b). Finally, γ1 MCP makes a few hydrogen bonds to Dec[P74-26] using its P-domain (Supplementary Fig. 10c).

## Discussion

The P74-26 Major Capsid Protein uses unique architectural features—lassos, rings, and flaps—to enhance stability of the capsid. We observe enhanced hydrophobic interactions at the subunit–subunit interfaces within the capsid. Hydrophobic interactions are estimated to be >2-fold higher for P74-26 than for other mesophilic homologs (Supplementary Tables 3 and 4). This observation can partially explain the enhanced thermostability of the P74-26 capsid, as the hydrophobic effect increases in strength at high temperature[32]. In contrast, we observe no significant alteration in the number of hydrogen bonds or salt bridges (Supplementary Tables 3 and 4), other interactions that have been seen to confer thermostability for some globular proteins[33–36].

We were not particularly surprised to find increased hydrophobic interactions in the P74-26 capsid. Numerous studies of thermophilic globular proteins show increased hydrophobic forces as a major contributor to thermal stability[33,37–40]. However, there are two things that make the P74-26 capsid a unique model system: (1) high internal pressure from tightly packaged DNA induces mechanical stress in the capsid[7,8,27] and (2) it is a self-assembling structure in which inter-subunit architecture and topology plays an important role in overall stability. We can derive these principles by comparing the P74-26 structure with those of numerous mesophilic homologs.

We find that the P74-26 capsid is stabilized by several loops and extensions that form topological linkages between subunits. The unique E-loop lasso attaches to the G-loop and P-domain of a neighboring MCP, which act as a hitching post for topologically tethering the lasso. Furthermore, the N- and C-arms, together with the Dec-arm, completely ring the E-loop β-strands, another architectural element unique to P74-26 (Fig. 5e). Thus, the E-loop is ringed toward the base and lassoed towards the tip.

A second lasso is formed by the N-arm, which forms both intra-capsomer and inter-capsomer interactions. The N-arm binds to the E-loop of a neighboring MCP within a capsomer through the upper-arm strand, the forearm helix, and hand region (Fig. 5a, c). Furthermore, the elbow, forearm, and hand stabilize capsomer–capsomer interactions by binding with Dec[P74-26] and an MCP subunit that lies across the twofold/quasi-twofold axes (Fig. 6a, b). While the N-arm is technically not a closed loop, P74-26 effectively closes the loop using the unique S-loop to fix the position of the forearm and hand regions (Fig. 6c). We find no similar N-arm lasso architecture in mesophilic Caudoviruses[31,41–46], which suggests that this architecture is important for enhancing capsid stability.

Although lassos are not found in other Caudoviruses, the distantly related herpesviruses contain analogous lasso architecture in the N-arm of the HK97 fold[47–49]. Much like the N-arm of P74-26, the herpesvirus N-arm lassos are not true closed loops.

Despite this seeming similarity, the herpesvirus lassos function differently. They exclusively stabilize inter-capsomer interactions, whereas the P74-26 N-arm lasso stabilizes both intra-capsomer and inter-capsomer interactions. Moreover, the pentons of herpesvirus capsids do not exhibit lasso interactions and the inter-actions are variable within hexon subunits, while P74-26 has nearly identical lasso interactions throughout both pentons and hexons. These observations indicate that the lasso architecture likely evolved independently and highlights the flexibility of topological stabilization mechanisms. We hypothesize that the extendable arm architecture facilitates evolution of stronger interactions within self-assembling systems such as capsids. These open-ended loops can be easily enhanced through serial single-residue extension. Perhaps this ease of evolution is why extended arm lassos are found in both P74-26 and herpesviruses. These extended lassos are similar to N- and C-terminal extensions that mediate assembly in other viruses (i.e. C-terminal extensions in SV40 capsid coat proteins[50]). We anticipate that these types of extended lassos can be useful for engineering more stable capsids and other self-assembling particles.

Another advantage of the lasso architecture is that it can adopt a less extended conformation. P74-26 MCP has two lassos on either end of the HK97 fold, both of which are presumably present in the much smaller procapsid. By using these lasso structures, P74-26 can retain high stability, while providing conformational flexibility to expand during maturation. We hypothesize that the lassos are less extended in the procapsid; upon capsid expansion, the lassos reach their full extension, where they lock into place. The full extension observed in the mature capsid would provide tensional integrity, as we discuss below.

P74-26 uses interweaved flaps to topologically stabilize inter-capsomer contacts. The T-loop stabilizes inter-capsomer interactions by inserting into a groove on the P-domain of an MCP subunit in a neighboring capsomer. These T-loop interactions are found ringing the threefold/quasi-threefold axes along the inside face of the capsid (Fig. 7b; Supplementary Fig. 7b). Similarly, the inter-capsomer twofold/quasi-twofold interactions are stabilized on the outside face of the capsid by the interleaved arrangement of the N-arms (Fig. 7c; Supplementary Fig. 7a). These overlapping structures resemble the interleaved arrangement of flaps in the top of a moving box. In this manner, the outside and inside faces of the capsid are stabilized by two separate interleaving flap inter-actions. We propose the moving box arrangements seen at symmetry and quasi-symmetry axes greatly strengthens the capsid against internal pressure because they are topologically challenging to disrupt. However, these arrangements are also presumably challenging to assemble, which raises the important question of how the P74-26 capsid assembles with an interleaved architecture.

The P74-26 Decoration protein likewise adopts a unique structural arrangement, contributing substantially to capsid thermostability. Decoration proteins increase capsid stability[28,29,51], although additional roles have been postulated[52]. The threefold/quasi-threefold axes are stabilized by the Dec[P74-26] trimer. Compared to mesophilic Caudoviruses, the Dec[P74-26] trimer interacts with more subunits across a much larger inter-action area (Fig. 8a). The total interaction area per Dec[P74-26] subunit is quite remarkable: ~4100.Å$^2$ for a 146-residue protein. Our previous study showed that Dec[P74-26] is substantially more stable than its mesophilic homologs, and this stabilization is primarily through formation of an extraordinarily tight trimer[23]. However, the trimerization interactions only account for a small fraction of the total Dec[P74-26] interaction area (~18% of the total interaction area per Dec[P74-26] subunit). This suggests that the Dec[P74-26] interaction with the capsid contributes a substantial amount of stability.

Interactions between Dec[P74-26] trimers forms a cage holding the capsid together (Fig. 8b). This arrangement is unique to P74-26. For example, phages lambda and TW1 use a very similar Decoration Protein fold[23], but the interaction of their Dec-arm with other capsid proteins is much more limited[29,44]. Further-more, the unrelated decoration protein of phage L does not connect with neighboring trimers, and in fact is missing at the quasi-threefold axes[52]. T4 phage is decorated with the Soc protein that interacts with neighboring Soc subunits at the threefold and quasi-threefold axes; however, Soc is present in relatively low occupancy (~50%), so the cage is incomplete[41]. Because decora-tion proteins are usually absent in the procapsid[29], we expect that the Dec[P74-26] cage would assemble cooperatively upon capsid expansion to stabilize the capsid. Future experiments will inter-rogate the role of cooperativity in assembly and stability.

The architectural enhancements in P74-26 MCP and Dec provide stabilization against high internal pressure. How do forces from internal pressure act on the capsid, and how does the capsid architecture resist these stresses? If we assume that the pressure from encapsulated DNA is distributed uniformly around the isometric capsid, then all points on the capsid experience a vector of force that is perpendicular to the surface of the capsid. By analogy, the capsid is experiencing forces that are similar to those of a balloon. Thus, the internal pressure causes lateral stresses on capsid interactions. While the high internal pressure exhibited by the phage challenges capsid stability, it may allow for stability mechanisms that rely on tensegrity. It is possible that the internal pressure can be harnessed to produce a particle stabilized by the stress on the individual subunits. In support of this, picornavirus capsids, which contend with considerably less internal pressure, can be stabilized by minor modifications to the capsid[53].

The architecture of the P74-26 capsid is constructed to with-stand lateral stress through tensional integrity. Tensional integ-rity, or tensegrity, is a generalized mechanism for architectural stability that involves structured regions held together by a net-work of flexible elements that are under continuous tension[54]. In the P74-26 capsid, the A- and P-domains are the structured regions, and the lassos and extended arms are the flexible ele-ments that transmit tension. For example, the E-loop lasso will become taut against the hitching post formed by the G-loop and the P-domain of the neighbor. Likewise, the N-arm forms a lasso whose end is held in place by the S-loop that locks into the groove between the forearm and hand (Fig. 6c). Thus, we predict that the S-loop will exhibit the hallmarks of a catch bond, a non-covalent bond that becomes stronger under tension[55]. Furthermore, the P74-26 capsid contains several flaps that interleave with each other. These interactions would topologically resist the lateral and longitudinal stresses of internal pressure. As a whole, these lasso and flap elements use tension to resist structural failure of the capsid. The tensegrity mechanism observed here is simply a more elaborate example of capsid tensegrity suggested by Caspar many years ago[56].

The lasso, flap, and arm interactions are positioned such that the internal pressure distributes the stress across multiple bonds. For example, the E-loop β-sheet experiences forces along the axis of the sheet. Thus, all bonds holding the sheet together are under stress rather than the orthogonal geometry in which the stress is just on the bonds at the end of the sheet. Capsid failure would require simultaneous disruption of many bonds (a shearing geometry), rather than an unzipping geometry in which the bonds rupture one at a time[57]. Pioneering single molecule studies have shown that a shearing geometry requires much higher forces to disrupt than when forces act in an unzipping geometry[58–60]. Thus, the P74-26 capsid is constructed such that lateral forces act in a shearing geometry, resulting in high tensegrity.

In addition to the unique stabilizing architecture of the capsid, P74-26 also adopts a non-canonical mechanism for altering capsid capacity. The capsid of P74-26 is larger than in most Caudoviruses, which correlates with its abnormally large genome. Most $T = 7$ Caudoviruses have genome sizes between 30 and 50 kb (Supplementary Table 2), while phage P74-26's genome is nearly twice as long at 83 kb[24]. Based on genome size we had predicted that the capsid would be $T = 12$ (average genome size ~80 kb[61]), although $T = 9$ or $T = 13$ would have been possible (average genome size ~70 or ~120 kb, respectively). The P74-26 capsid attains this larger size by significantly increasing the capsomer size rather than changing the icosahedral complexity. The capsomer is larger because the P74-26 MCP covers more surface area than normal, despite a typical length for MCP. Consequently, the capsomer is slightly thinner than normal (Fig. 4b). Thus, the number of residues in MCP does not predict total area covered, and genome size does not predict triangulation number.

Recently, Bayfield et al.[62] determined the structure of a closely related $T = 7$ thermophilic phage, which similarly uses enlarged capsomers to increase capsid capacity. To our knowledge, this is a non-canonical mechanism for increasing capsid capacity. There are two classic mechanisms for enlarging a capsid: (1) increasing the triangulation number and (2) conversion of an isometric to prolate head. In the first case, hexons are added across all faces of the capsid, whereas in the second, hexons are added across ten of the icosahedral faces such that the capsid is elongated in one dimension (Fig. 9). In both cases, the capsomers remain the same size. Here, we have identified a third mechanism for evolution of a larger capsid: increasing the size of the capsomer.

These three mechanisms have very different evolutionary barriers. The two classical mechanisms can be implemented through simple mutations and have been observed numerous times. In many viruses, simple point mutations modify the triangulation number[5,63,64]. Furthermore, the triangulation number of some capsids can be altered without changing MCP sequence[3,65,66]. Likewise, single point mutations in T4 phage convert the capsid from prolate to isometric or generate giant heads in which the long axis of the prolate head is lengthened[4,67]. Therefore, the evolutionary barriers for altering capsid volume through the two classical mechanisms appear to be quite low. In contrast, the enlarged capsomer strategy identified here requires multiple, extensive alterations to the sequences of capsid proteins. The larger P74-26 capsid necessitates large changes to the eight separate modifications to the MCP structure, as well as the Dec-arm (Figs. 3a, b and 4c, d). This begs the question: why did phage P74-26 utilize this seemingly more challenging evolutionary strategy rather than the easier, classical strategies? What

constraints prevented evolution of a larger capsid through the classical routes?

Our first hypothesis is that the lassos, flaps, and arms stabilizing the P74-26 capsid require a larger capsomer for function. It is possible that the lassos need extra space to open enough for a hitching post structure to insert. Likewise, the flaps and arms may require a certain length to elicit their stabilizing activity. If this were the case, then the architectural elements stabilizing the capsid require larger than normal capsomers. In this scenario, the larger capsomer is the selected structural feature and the $T = 7$ geometry is a spandrel: a biological structure that is a byproduct of evolution rather than a result of direct selection[68]. However, we do not favor this hypothesis because lassos are found in herpesviruses, in which the HK97 fold is a typical size (Herpesvirus MCPs have several other tower domains that increase size, but these domains are not part of the main capsid floor and do not contribute to the HK97 fold[49]). Furthermore, other Caudovirus MCPs contain long N-arms (e.g. Sf6 phage[31]) or E-loops that are opened nearly as wide as P74-26 (e.g. P22 phage[43]), but these proteins are of typical size. Nonetheless, this hypothesis remains untested.

A second hypothesis is that the genome size and capsid capacity coevolved through small concomitant increases. If the ancestral phage evolved a slightly larger genome than can be accommodated in the capsid, then there may be selective pressure for a slightly larger capsid. Increasing T number or transition to a prolate head substantially increases the capsid volume, resulting in large drops in internal pressure. These transitions may be discouraged because internal pressure needs to be maintained for infection[26]. To avoid large changes in internal pressure, larger capsomers may slowly coevolve with a larger genome.

Our final hypothesis is that capsid geometry has a direct effect on overall capsid stability. We hypothesize that the $T = 7$ geometry is inherently more stable than higher triangulation numbers due to variable conformations of hexons. All capsids that are $T = 9$ or higher have more than one type of hexon present, while all capsids $T \leq 7$ have exactly one type of hexon (except $T = 1$, which has no hexons[69,70]). For example, $T = 7$ has a single-pucker hexon conformation, while $T = 9$ has both winged and flat hexons (Supplementary Fig. 11a, b). We also note that prolate capsids have multiple types of hexons (generally three or more hexon conformations; Fig. 9). Thus, major capsid proteins in $T > 7$ viruses must accommodate the hexon conformational heterogeneity, which may adversely affect stability.

We hypothesize that $T = 7$ geometry is the highest complexity (i.e. largest size) that is inherently stable. More complex geometries would introduce instability through variation in hexon conformation. This inherent instability may require extra stabilization mechanisms to mitigate, such as decoration proteins to cement the structure in place. We envision two non-mutually exclusive disadvantages of $T > 7$ geometry. First, each of the separate hexon conformations must remain functional and stable, which would constrain evolution of MCP proteins for greater stability. The second benefit is that a lower triangulation number results in fewer subunit–subunit interfaces, thus minimizing the number of weak points in the capsid. In support of this hypothesis, the extremophilic, archaeal virus HSTV-2 (*Halorubrum sodomense* tailed virus 2) packages its ~68 kb genome into a $T = 7$ head[71]. HSTV-2 utilizes a larger-than-normal capsid, and also has trimeric decoration proteins that sit at the threefold/quasi-threefold axes. The fact that this mechanism for capsid enlargement has only been observed in extremophiles supports the idea that $T = 7$ geometry has a beneficial effect on stability. In further support of our hypothesis, all known $T > 7$ capsids use decoration proteins (to our knowledge), while many $T = 7$ viruses lack decoration proteins.

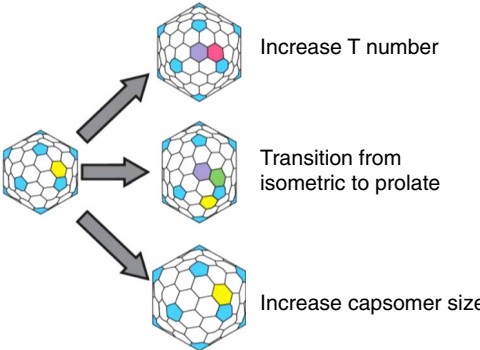

**Fig. 9** Mechanisms for increasing capsid capacity. P74-26 adopts a novel mechanism for enhancing capsid capacity by increasing the size of capsomers, while retaining $T = 7$ geometry

If various triangulation numbers have different inherent stability, this would suggest that each geometry exhibits weak points at different regions of the capsid, as has been predicted from theoretical work[72]. We hypothesize that the threefold/quasi-threefold axes represent the weak points in a $T = 7$ lattice. In support of this hypothesis, decoration proteins of $T = 7$ Caudoviruses are commonly found at the threefold/quasi-threefold axes (Supplementary Table 2)[29,44,52]. Furthermore, these axes are stabilized by covalent cross-links in HK97 phage[45] and T-loop flaps in P74-26 (Fig. 7b). To examine this idea further, we note that $T = 9$ phage also use decoration proteins at the threefold axes[73,74], while $T = 12$ and $T = 13$ phage use decoration proteins at the centers of capsomers[61,75,76].

We note that all of our analysis has primarily focused on Caudoviruses. These viruses do not generally break down their capsids as part of their lifecycle, so the capsid has no selective pressure to be labile. In fact, the high pressure of packaged DNA presents a high selective pressure to evolve stable capsids. It is likely that other types of viruses use different stability mechanisms, particularly viruses that disassemble their capsids as a necessary part of their lifecycle.

## Methods

**Growth/purification of P74-26 virions.** P74-26 phage was propagated in the host strain *T. thermophilus* HB8 (ATCC 27634) using fresh overnight cultures grown at 65 °C in Thermus growth medium (4 g L$^{-1}$ yeast extract, 8 g L$^{-1}$ tryptone, 3 g L$^{-1}$ NaCl, 1 mM MgCl$_2$, 0.5 mM CaCl$_2$). Four milliliters of P74-26 phage stock at $1 \times 10^6$ plaque-forming units per mL (PFU mL$^{-1}$) was combined with 6 mL of fresh *T. thermophilus* HB8 and incubated at 65 °C for 10 min. Reaction mixture was then inoculated into 1 L of Thermus Medium, and incubated at 65 °C while shaking for 4–5 h. Lysate was then spun at $4000 \times g$ for 20 min, and supernatant was incubated with DNase I (2 Units mL$^{-1}$) and RNase A (1 μg mL$^{-1}$) for 1 h at 30 °C. To precipitate virions, solid NaCl was added to 1 M final concentration and Polyethylene Glycol MW 8000 was added to a final concentration of 10% (w/v) while stirring. Phage stock was then incubated on ice overnight. The next day, precipitated phage stock was spun at $11,000 \times g$ for 20 min at 4 °C. The resulting phage pellet was resuspended in 2 mL of 50 mM Tris pH 7.5, 100 mM NaCl, and 1 mM MgSO$_4$. Resuspension was supplemented with 0.4 g solid CsCl and added to a CsCl step gradient (2 mL steps each of 1.2, 1.3, 1.4, 1.5 g mL$^{-1}$ CsCl and 1 mL cushion of 1.7 g mL$^{-1}$ CsCl, in 50 mM Tris pH 7.5, 100 mM NaCl, 1 mM MgSO$_4$). Gradients were prepared in 12 mL ultracentrifuge tubes and spun in a Beckman SW40-Ti rotor at 38,000 RPM for 18 h at 4 °C. Sedimented virions were isolated and dialyzed twice overnight into 2 L of 50 mM Tris pH 8.0, 10 mM NaCl, 10 mM MgCl$_2$ at 4 °C. P74-26 virions were then concentrated to $1 \times 10^{12}$ PFU mL$^{-1}$ for subsequent use in electron microscopy.

**Thermal stability assay.** Purified P74-26 virions were diluted in sample buffer (50 mM Tris pH 8.0, 10 mM NaCl, 10 mM MgCl$_2$) to a final concentration of ~1 × 10$^8$ PFU mL$^{-1}$ and incubated at 80 °C for 0, 0.5, 1, 2, 4, or 6 h in triplicate. Incubated virions were serially diluted and 100 μL aliquots of several dilutions were each mixed with 150 μL of fresh *T. thermophilus* HB8 (OD$_{600}$ = 1.0). The adsorption reactions were then incubated at 65 °C for 10 min and mixed with 2.5 mL molten Thermus Medium supplemented with 0.75% agar. The molten reaction mixture was then spread evenly on top of TR Gellan Gum plates (2 g L$^{-1}$ yeast extract, 4 g L$^{-1}$ tryptone, 1 g L$^{-1}$ NaCl, 15 g L$^{-1}$ Gellan gum, 1.5 mM MgCl$_2$, 1.5 mM CaCl$_2$, pH 7.5). Once the soft agar solidified, plates were inverted and incubated overnight at 65 °C yielding a bacterial lawn with discernible phage plaques. Plaques were then counted from three plates for each incubation.

**Negative-staining EM.** Infections of *Thermus thermophilus* HB8 were performed as described above (growth/purification of P74-26 virions). Aliquots of the infection culture were diluted 1:1 in Thermus growth medium and applied to carbon-coated 400-mesh grids 1 h after infection. Excess sample was then blotted from the grid surface, and grids were finally stained with 1% uranyl acetate. Data were then collected on a 120 kV Philips CM-120 microscope fitted with a Gatan Orius SC1000 detector.

**Cryo-EM specimen preparation.** 400-mesh copper lacey carbon support film grids (Electron Microscopy Sciences) were glow discharged using a Pelco easiGlow (Pelco) for 45 s at 20 mA (negative polarity) prior to sample application. In all, 3.5 μL of P74-26 virions at $1 \times 10^{12}$ PFU mL$^{-1}$ was applied to a cleaned grid at 22 °C and 90% humidity in a Vitrobot Mark IV (FEI). Samples were then blotted for 8 s after a wait time of 15 s. Sample-coated grids were then vitrified by plunging into liquid ethane. Prepared grids were then stored submerged in liquid nitrogen prior to data collection.

**Data collection.** Micrographs were collected on a 300 kV Titan Krios electron microscope (FEI) equipped with a K2 Summit direct electron detector (Gatan). Images were collected at a magnification of ×130,000 in super-resolution mode at a pixel size of 0.529 Å per pixel and a total dose of 48e$^-$ Å$^{-2}$ per micrograph. Micrographs were collected with a target defocus of −0.2 to −1.2 μm. For each micrograph, 32 frames were motion-corrected using the *Align Frames* module in IMOD (http://bio3d.colorado.edu/imod/betaDoc/man/alignframes.html) with 2× binning yielding a final pixel size of 1.059 Å per pixel. In total, 4611 micrographs were obtained from a single 3-day collection.

**Data processing.** CTFFind4 within the *cis*TEM software package[77] was used to determine defocus values for each resulting micrograph. An initial particle set of 37,046 particles was then picked using the *Find Particles* action in *cis*TEM with a specified maximum particle radius of 390 Å. This led to selection primarily of P74-26 capsids without selecting virion tail structures. The resulting particle stack was extracted using a box size of 1024 pixels, and then sorted into 20 classes using 20 cycles of 2D classification in *cis*TEM. Particles from 2D class averages containing complete icosahedral capsids were extracted, yielding a stack of 28,880 particles. The particle stack was then used to generate an ab initio model for subsequent refinement using the Ab initio 3D action in *cis*TEM. Finally, 4× binned, 2× binned, and unbinned particle stacks, and particle parameters were exported to the Frealign software package (Frealign version 9.11 (ref. [78])).

To speed up initial data processing, the 4× binned particle stack was used for five cycles of global alignment (mode 3) with a resolution range of 300–12 Å in Frealign using the ab initio-generated model as a template. The 2× binned particle stack was then subjected to 12 rounds of refinement (mode 1) in Frealign, increasing the high-resolution limit each three cycles from 12 to 6 Å. The 2× binned particle stack was then subjected to 20 rounds of 3D classification using four total classes and a high-resolution limit of 6 Å. Two classes containing high-quality particles were combined using *merge_classes.exe* in Frealign, yielding a particle substack containing 23,178 particles. The particle substack (1.059 Å/pixel) was then subjected to eight cycles of refinement and reconstruction in Frealign mode 1, yielding a resolution of 3.3 Å (0.143 FSC cutoff). FSC curves were calculated in Frealign using odd and even particle half-sets. The map was sharpened using automatically calculated B-factors (−98 Å$^2$) using *bfactor.exe* in Frealign. The sharpened map was used for preliminary model building and structure refinement. The final cycle of refinement included Ewald sphere curvature corrected reconstructions of the same and opposite handedness by setting IEWALD in Frealign to either 1 or −1, respectively. The Ewald sphere curvature corrected reconstruction of the same handedness yielded the final map with a resolution of 2.8 Å (0.143 FSC cutoff; 3.1 Å at 0.5 FSC cutoff). The final map was sharpened with a B-factor of −100 Å using *bfactor.exe* in Frealign. The final map was then used for final model building.

**Model building.** For the P74-26 MCP atomic model, a poly-alanine chain was initially built into the density map and manually refined using Coot[79]. Then, side chains were assigned, and a complete model of MCP was built and refined manually. For the P74-26 decoration protein, the previously determined atomic model of the decoration protein trimer (PDB: 6BL5) was rigid-body fit into a corresponding portion of the density map. The N-terminal 16 residues of the decoration protein were then built into the density map, and the protein trimer was refined manually in Coot. The models of the MCP monomer and decoration protein trimer were then refined into the P74-26 density map using the Phenix real-space refine procedure[80]. These models were then used to generate the P74-26 icosahedral asymmetric unit. The MCP P- and A-domains were first rigid-body fit into the density corresponding to each MCP subunit of the asymmetric unit (six hexon subunits and one penton subunit). Then, the N-arm, C-arm, and E-loop of each subunit were individually rigid-body fit into the corresponding density to account for variability in the orientation of these loops throughout the asymmetric unit. The refined decoration protein model was then fit into the corresponding density of the asymmetric unit, comprising two decoration protein trimers and an additional monomer. The complete asymmetric unit was then manually refined in Coot, and final real-space refinement of the asymmetric model was performed using Phenix. The final real-space refinement consisted of 10 cycles of refinement using the 2.8-Å Ewald sphere corrected map as input. The real-space refinement statistics are listed in Supplementary Table 1.

**Reporting summary.** Further information on research design is available in the Nature Research Reporting Summary linked to this article.

## Data availability

Model coordinates of the P74-26 capsid asymmetric unit were deposited to the Protein Data Bank (PDB: 6O3H). The density map of the icosahedral capsid reconstruction was deposited to the EM Data Bank (EMD-0618). The source data underlying Fig. 1B are provided as a Source Data file.

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

## Acknowledgements

The authors thank Drs. C. Xu, KK Song, and K. Lee for assistance with data collection, and Drs. C. Xu, A. Korostelev, C. Gaubitz, M. Herzik, Ms. J. Hayes, and A. Jecrois for advice on data processing. We thank members of the Kelch, Royer, and Schiffer labs for helpful discussions. We thank C. Gaubitz, J. Hayes, and J. Magrino for critical reading of the manuscript. This work was funded by the Pew Charitable Trusts and the National Science Foundation (grant number 1817338).

## Author contributions

N.P.S. and B.A.K. designed research; N.P.S. and E.A. performed research; N.P.S., G.D. and B.A.K. analyzed data; and N.P.S. and B.A.K. wrote the paper.

## Competing interests

The authors declare no competing interests.
