## [Peer Review File · Nature Communications]

Reviewers' Comments:

Reviewer #1:

Remarks to the Author:

NCOMMS-19-03838

"Principles for enhancing virus capsid capacity and stability from a thermophilic virus capsid structure" by Prof Kelch and colleagues

In this manuscript, Kelch and colleagues report the cryoEM structure of the thermophilic phage P74-26. The capsid subunits have the HK97 fold. The major finding of the paper is that the capsid has evolved a large size to contain its genome, but not by adding additional subunits as seen for mesophilic phages. Rather, the capsid is a T=7 structure, which typically can encapsidate around 45 kb of DNA. The subunits themselves are bigger than usual for the HK97 fold by extension of helices in the A-domain and a longer E-loop. Many additional small loops are also involved in inter- and intra-capsomer stabilization. Thus, the virus has evolved an interesting solution to remaining viable in harsh growing conditions.

A real positive of the manuscript is the beautiful figures that Kelch and colleagues use to explain the differences between their capsid subunit and the typical subunit with the HK97 fold. My quibble with the manuscript is not about the science but the presentation. Much of the results describes the endless loops and lassos and latches and hitches, which could be summed up in a table. The middle school dance analogy is just over the top. The casual nature of the descriptions is simply too much, and should be made more formal and much shorter.

Other comments:

The authors make no mention of how scaffolding proteins are involved in capsid size determination. The Dec protein does not drive capsid size. Presumably this phage has a scaffolding protein.

A structure of a very related phage was just published from the Anston group (Proc Natl Acad Sci U S A. 2019 Feb 26;116(9):3556-3561. doi: 10.1073/pnas.1813204116). I highly recommend harmonization of the naming of the loops and other structural features so the same terms are used in both manuscripts.

Please call it the HK97 fold, not the Johnson fold. It has been this way in the literature for 20 years. If one does a pubmed search for 'johnson fold', the results are all from authors with the last name Johnson.

The discussion of T=7 virus capsid structure is interesting. I'm not certain one can make the conclusions the authors suggest based on one example. Hyperthermophilic phages have evolved an amazing variety of solutions to remain infectious at high temperatures, as clearly shown by David Prangishvili and others. The authors should consider softening these conclusions.

Reviewer #2:

Remarks to the Author:

Stone et al, 'Principles for enhancing virus capsid capacity and stability from a thermophilic virus capsid structure' presents the high resolution structure of a thermostable phage: P74-26. The authors correlate the unusual features of the structure with the thermostability of the particles. This is an elegant piece of work, and the structure is beautiful, with the experimental results supporting the structural interpretations. The paper is well written and the figures are mostly

well executed and clear. Based on the numerous unusual features of the structure I think the paper is of sufficient interest to warrant publication in Nature Communications, and I find most of the explanations very sensible (eg the discussion of "tensegrity", use of shearing forces etc). I do however have a number of queries/suggestions.

Summary, final sentence: I find this plausible but I am not sure there is evidence? There is discussion later in the paper but the evidence is circumstantial and the sentence should be softened.

Introduction line 4 (and elsewhere): A problem with case of reference library entry.

Results, thermostability: Marked thermal stability can be achieved in obscure ways – for instance hepatitis A empty and full particles are stable up to 80 °C – far higher (~30 °C higher) than most other picornaviruses, but there are no stand-out structural differences (Wang et al 2015). In many other viruses thermal stability is modulated by very minor sequence changes (stability often varying significantly between different field isolates). It is conceivable that the addition of genome pressure in phages adds to the stress to force more radical solutions, but there would need to be clear evidence for this.

Page 3. Please provide evidence of density quality in a main figure - after all it is nice and underpins the whole paper.

Page 3. "Thus, P74-26 has presumably evolved a bigger capsid to hold its large genome". Well, a larger capsid might have encouraged a larger genome to maintain the pressure to eject the genome. And a larger capsid can eject more DNA for a given pressure drop, to power DNA through the very long tail. So for me it's not clear what the 'driver' is.

Page 3: "The P74-26 capsomers are also less curved than in typical Caudoviruses (Fig. 3B; Fig. S4C)". I am not convinced ... in Fig 3A the P74-26 particles look more spherical.

Page 4, Fig 4A: Labelling is a bit clunky and maybe colouring could be better? For instance better to label E-loop on the far end of the loop not the other end of the strand.

Page 6: "The N-arm forms a 'double-lasso' structure that stabilizes the assembly at the two-fold and quasi-two-fold axes of symmetry between capsomers (Fig 6C)". Perhaps it was lasso-fatigue but I found the pictures really not very clear – the information is there but the features don't stand out at all, please redraw.

Page 6: I can't see in any of the main pictures a clear rendition of the fold of Dec. I suspect that this is because of the prior crystal structure, but for completeness I feel the actual structure in the virus must be presented in a main figure.

Page 6 (or thereabouts): I failed to find any explanation of the penton/hexon switch and a consideration of the overall assembly pathway. Since the structure is of the final mature particle perhaps this is not possible, but I would like to see more consideration of the penton/hexon changes.

Page 8: "...the lassos reach their full extension, where they lock into place. The full extension observed in the mature capsid would provide tensional integrity, as we discuss below". In some ways the lasso is a more robust/versatile version of the N and C terminal extensions that mediate assembly in very many viruses.... eg the extended SV40 C-terminal arms.

Page 9: "middle-school slow dance". I don't mind Americanisms, but for non US readers this is really obscure.

Page 10, towards end of page: the discussion assumes that the role of the decoration proteins is to increase strength? Is this established?

Page 11: "We anticipate that these principles may be modified for viruses that disassemble their capsids as a necessary part of their lifecycle. Future research will delineate the differences between the two classes of viruses." perhaps this is a bit simplistic - there are other viruses with genomes that generate pressure (eg dsRNA viruses), and have very different solutions.

Thermal stability assay: would it not be worth looking to see what the breakdown products are - to really see where the weak points are?

Page 12: "The unbinned particle substack (1.059 Å/pixel) was then..." - this is confusing, it is actually 2x binned from original pixel size.

Reviewer #1 (Remarks to the Author):

NCOMMS-19-03838

"Principles for enhancing virus capsid capacity and stability from a thermophilic virus capsid structure" by Prof Kelch and colleagues

In this manuscript, Kelch and colleagues report the cryoEM structure of the thermophilic phage P74-26. The capsid subunits have the HK97 fold. The major finding of the paper is that the capsid has evolved a large size to contain its genome, but not by adding additional subunits as seen for mesophilic phages. Rather, the capsid is a T=7 structure, which typically can encapsidate around 45 kb of DNA. The subunits themselves are bigger than usual for the HK97 fold by extension of helices in the A-domain and a longer E-loop. Many additional small loops are also involved in inter- and intra-capsomer stabilization. Thus, the virus has evolved an interesting solution to remaining viable in harsh growing conditions.

A real positive of the manuscript is the beautiful figures that Kelch and colleagues use to explain the differences between their capsid subunit and the typical subunit with the HK97 fold. My quibble with the manuscript is not about the science but the presentation. Much of the results describes the endless loops and lassos and latches and hitches, which could be summed up in a table. The middle school dance analogy is just over the top. The casual nature of the descriptions is simply too much, and should be made more formal and much shorter.

-We appreciate the reviewer's comments on our manuscript and agree that the presentation can be improved. We have revised the text to remove the 'middle-school dance' analogy and made the descriptions more formal. We have also abbreviated the Discussion and removed the 'Conclusions' section.

Other comments:

The authors make no mention of how scaffolding proteins are involved in capsid size determination. The Dec protein does not drive capsid size. Presumably this phage has a scaffolding protein.

-The reviewer is correct in their assumption that phage P74-26 has a scaffolding protein. We now describe the role of Scaffolding protein in capsid size determination in the introduction (page 1, column 2).

A structure of a very related phage was just published from the Anston group (Proc Natl Acad Sci U S A. 2019 Feb 26;116(9):3556-3561. doi: 10.1073/pnas.1813204116). I highly recommend harmonization of the naming of the loops and other structural features so the same terms are used in both manuscripts.

-The nomenclature for the Major Capsid Protein structural features has been revised to coincide with that used by Bayfield, et al. We have changed the names of the "F'-loop" and "P-domain latch" to "T-loop" and "S-loop", respectively to avoid confusion.

Please call it the HK97 fold, not the Johnson fold. It has been this way in the literature for 20 years. If one does a pubmed search for 'johnson fold', the results are all from authors with the last name Johnson.

-We have revised the text, changing all instances of "Johnson fold" to the more commonly used "HK97 fold".

The discussion of T=7 virus capsid structure is interesting. I'm not certain one can make the conclusions the authors suggest based on one example. Hyperthermophilic phages have evolved an amazing variety of solutions to remain infectious at high temperatures, as clearly shown by David Prangishvili and others. The authors should consider softening these conclusions.

-We agree with the reviewer that this putative stability mechanism is only one mechanism among many others. We have revised the Discussion to more accurately reflect this (page 6, column 2).

Reviewer #2 (Remarks to the Author):

Stone et al, 'Principles for enhancing virus capsid capacity and stability from a thermophilic virus capsid structure' presents the high resolution structure of a thermostable phage: P74-26. The authors correlate the unusual features of the structure with the thermostability of the particles. This is an elegant piece of work, and the structure is beautiful, with the experimental results supporting the structural interpretations. The paper is well written and the figures are mostly well executed and clear. Based on the numerous unusual features of the structure I think the paper is of sufficient interest to warrant publication in Nature Communications, and I find most of the explanations very sensible (eg the discussion of "tensegrity", use of shearing forces etc). I do however have a number of queries/suggestions.

-We thank the reviewer for their insights and positive comments.

Summary, final sentence: I find this plausible but I am not sure there is evidence? There is discussion later in the paper but the evidence is circumstantial and the sentence should be softened.

The final sentence of the summary/abstract has been changed to explain that the inherent stability of T=7 geometry is merely a hypothesis.

Introduction line 4 (and elsewhere): A problem with case of reference library entry.

-The references have been reformatted in numerical order as requested by Nature Communications. As such, all issues with reference calls in the text have been resolved.

Results, thermostability: Marked thermal stability can be achieved in obscure ways – for instance hepatitis A empty and full particles are stable up to 80 °C – far higher (~30 °C higher) than most other picornaviruses, but there are no stand-out structural differences (Wang et al 2015). In many other viruses thermal stability is modulated by very minor sequence changes (stability often varying significantly between different field isolates). It is conceivable that the addition of genome pressure in phages adds to the stress to force more radical solutions, but there would need to be clear evidence for this.

-We appreciate the reviewer's insight on thermal stability in other viruses. We have added a brief section to the Discussion to include this observation (page 5, column 2).

Page 3. Please provide evidence of density quality in a main figure - after all it is nice and underpins the whole paper.

-We agree with the reviewer that the quality of the density should be reflected in the main paper rather than the supplementary information. Accordingly, we have added panel c to figure 2 which shows the P-domain spine helix of the Major Capsid Protein modeled into the cryoEM density map.

Page 3. “Thus, P74-26 has presumably evolved a bigger capsid to hold its large genome”. Well, a larger capsid might have encouraged a larger genome to maintain the pressure to eject the genome. And a larger capsid can eject more DNA for a given pressure drop, to power DNA through the very long tail. So for me it’s not clear what the ‘driver’ is.

- We agree with the reviewer that the evolutionary driver is unclear. Therefore, we refer only to coevolution of capsid capacity and genome size. We have revised the Discussion to reflect the possibility of other evolutionary mechanisms (page 6, column 2).

Page 3: “The P74-26 capsomers are also less curved than in typical Caudoviruses (Fig. 3B; Fig. S4C)”. I am not convinced ... in Fig 3A the P74-26 particles look more spherical.

-While the P74-26 capsid particle does indeed have a higher “spherical factor” than HK97 or Lambda capsids, the P74-26 capsomers (Hexons are shown in figure 3B) cover a larger surface area than HK97 and are accordingly less concave. We have revised the results section to expand upon this observation (page 2, column 2).

Page 4, Fig 4A: Labelling is a bit clunky and maybe colouring could be better? For instance better to label E-loop on the far end of the loop not the other end of the strand.

-The labels and arrows used in the schematic (now Figure 5a) have been revised for clarity.

Page 6: “The N-arm forms a ‘double-lasso’ structure that stabilizes the assembly at the two-fold and quasi-two-fold axes of symmetry between capsomers (Fig 6C)”. Perhaps it was lasso-fatigue but I found the pictures really not very clear – the information is there but the features don’t stand out at all, please redraw.

-We have revised this figure panel (now Figure 7c) to enhance clarity. Additionally, we reference Supplementary Figure 7 panel a, which shows an annotated schematic of the 2-fold lasso interaction (page 5, column 1).

Page 6: I can’t see in any of the main pictures a clear rendition of the fold of Dec. I suspect that this is because of the prior crystal structure, but for completeness I feel the actual structure in the virus must be presented in a main figure.

-We agree with the reviewer that a clear rendition of the Dec protein should be included in the main manuscript. We have added panel “a” in the revised Figure 3 to reflect this inclusion.

Page 6 (or thereabouts): I failed to find any explanation of the penton/hexon switch and a consideration of the overall assembly pathway. Since the structure is of the final mature particle perhaps this is not possible, but I would like to see more consideration of the penton/hexon changes.

-Given that this work only encompasses the structure of the mature virion, it would be difficult to draw any meaningful conclusions regarding the assembly pathway. The manuscript includes a brief discussion of the conformational differences between hexons and pentons within the mature virion (page 2, column 2).

Page 8: "...the lassos reach their full extension, where they lock into place. The full extension observed in the mature capsid would provide tensional integrity, as we discuss below". In some ways the lasso is a more robust/versatile version of the N and C terminal extensions that mediate assembly in very many viruses.... eg the extended SV40 C-terminal arms.

-We agree with the reviewer that the extended lassos in P74-26 are more robust versions of N- and C-terminal extensions that mediate assembly in other viruses and address this in more detail in the Discussion (page 5, column 1).

Page 9: "middle-school slow dance". I don't mind Americanisms, but for non US readers this is really obscure.

-We agree and the updated text reflects these changes.

Page 10, towards end of page: the discussion assumes that the role of the decoration proteins is to increase strength? Is this established?

-We have now addressed this point in the Discussion section (page 5, column 1). Briefly, the only known role for decoration proteins is to enhance capsid stability.

Page 11: "We anticipate that these principles may be modified for viruses that disassemble their capsids as a necessary part of their lifecycle. Future research will delineate the differences between the two classes of viruses." perhaps this is a bit simplistic - there are other viruses with genomes that generate pressure (eg dsRNA viruses), and have very different solutions.

-We agree with the reviewer that this portion of the text was oversimplified, and we have revised the text to avoid confusion (page 7, column 1).

Thermal stability assay: would it not be worth looking to see what the breakdown products are - to really see where the weak points are?

-We thank the reviewer for their suggestion. Accordingly, we have included additional electron micrographs to reflect the state of particles following incubation at 80°C (see Supplementary Figure 1; in text: page 2, column 1).

Page 12: "The unbinned particle substack (1.059 Å/pixel) was then..." – this is confusing, it is actually 2x binned from original pixel size.

-The text has been revised to avoid confusion among the different particle stacks used for refinement of the data (page 8, column 1).

Reviewers' Comments:

Reviewer #2:

Remarks to the Author:

The authors have addressed the points from the referees in a careful way.

The manuscript is still rather long and discursive, but since my points have been answered I am happy to support publication.

Reviewers' Comments:

Reviewer #2:

Remarks to the Author:

The authors have addressed the points from the referees in a careful way.

The manuscript is still rather long and discursive, but since my points have been answered I am happy to support publication.

We thank the reviewer for their comment, and appreciate their support for our manuscript.